# The Landscape of Circular RNAs in Cardiovascular Diseases

**DOI:** 10.3390/ijms24054571

**Published:** 2023-02-26

**Authors:** Qi Long, Bingjie Lv, Shijiu Jiang, Jibin Lin

**Affiliations:** 1Department of Cardiology, Union Hospital, Tongji Medical College, Huazhong University of Science and Technology, Wuhan 430022, China; 2Hubei Key Laboratory of Biological Targeted Therapy, Union Hospital, Tongji Medical College, Huazhong University of Science and Technology, Wuhan 430022, China; 3Hubei Provincial Engineering Research Center of Immunological Diagnosis and Therapy for Cardiovascular Diseases, Union Hospital, Tongji Medical College, Huazhong University of Science and Technology, Wuhan 430022, China

**Keywords:** circRNA, cardiovascular disease, biomarker

## Abstract

Cardiovascular disease (CVD) remains the leading cause of mortality globally. Circular RNAs (circRNAs) have attracted extensive attention for their roles in the physiological and pathological processes of various cardiovascular diseases (CVDs). In this review, we briefly describe the current understanding of circRNA biogenesis and functions and summarize recent significant findings regarding the roles of circRNAs in CVDs. These results provide a new theoretical basis for diagnosing and treating CVDs.

## 1. Introduction

The incidence of cardiovascular disease (CVD) has been increasing rapidly in recent years, which is the leading cause of death worldwide [1,2]. CVDs are characterized by high morbidity, high disability rate, and high mortality, as well as various complications. As such, there is an urgent need to identify new potential biomarkers and therapeutic targets for the prevention and treatment of CVDs.

The accumulating evidence has indicated that only a small percentage of the human genome encoded proteins and the rest were non-coding RNAs (ncRNAs) [3]. Recent studies have shown the roles ncRNAs played in cellular homeostasis and disease pathophysiology. Based on their size, ncRNAs could be divided into two groups: small ncRNAs and long ncRNAs. Circular RNA(CircRNA), a particular type of long ncRNA, forms a closed-loop framework by covalently constituting single-stranded RNAs without the usual terminal structures such as the 5′cap or polyadenylated tail. CircRNAs are also characterized by their high abundance and sometimes could even exceed 10 times that of linear transcripts [4].

In the past decade, great attention has been paid to the biogenesis and function of circRNAs in different diseases, including in CVDs. In addition, the potential clinical applications, such as diagnostic and prognostic biomarkers, as well as therapeutic targets in heart disease, are new emerging domains [5,6,7,8]. In the present review, the literature on circRNAs and CVDs published in the English language in PubMed data, from January 2013 to January 2023, was searched with keywords including “RNA, Circular” and “Cardiovascular Diseases”. We summarized the existing knowledge of circRNAs in CVDs, from cardiovascular-related cells to different diseases, and pointed out directions of potential future researches.

## 2. Biogenesis and Functions of circRNAs

CircRNAs are closed circular biomolecules, which distinguish them from other linear RNA biomolecules. They are circularized by joining the 3′ and 5′ ends together via exon circularization or intron circularization [9]. At present, three major mechanisms have been reported for the generation of circRNAs: “lariat-driven circularization”, “intron-pairing-driven circularization” and “RBP-driven circularization” [10] (Figure 1. Lariat-driven circularization is associated with exon skipping, in which one or more transcript exons are skipped, and then the remaining lariat itself is joined by a spliceosome and becomes an exon circle. Intron-pairing-driven circularization, on the other hand, is mostly related to complementary motifs present in the intronic regions. In this model, direct RNA base pairing with reverse complementary sequences, such as Alu repeats in the human genome across introns flanking exons, is brought into proximity to promote circularization [11,12]. CircRNAs can also be cyclized by RNA-binding protein (RBP). In RBP-driven circularization, trans-acting factors recognize and dock on specific motifs located in the introns flanking the circularized exons. The splice sites are brought into close proximity through these protein–protein interactions or dimerization, and after that the spliceosome could engage in a back-splicing reaction [13]. It is worth noting that circular intronic circRNAs (ciRNAs) have a special lariat circularization method in which only introns remain. CircRNAs can be classified into at least three categories: EcircRNAs (exonic circRNAs), EIcircRNAs (exonic-intronic circRNAs) and ciRNAs (circular intronic circRNAs), according to their synthesis mechanism and constituent components (Table 1) [14,15,16,17,18,19]. EcircRNAs now make up a notable proportion of the known circRNAs. 

Despite the rapid research progress in the field, the biological functions of circRNAs in eukaryotic cells have not been fully understood. To date, several potential functions of circRNAs have been revealed: (1) acting as miRNA sponges; (2) interacting with RNA-binding proteins (RBP); (3) acting as dynamic scaffolding molecules that modulate protein–protein interactions; (4) acting as transcription or translation regulators; (5) participating in the translation of proteins [13,20,21,22] (Figure 1). The function CircRNAs as miRNA sponge has been a focus of recent research. CircRNAs can competitively bind to miRNAs and lead to the reduction of miRNAs. Then, the reduced miRNAs have reduced inhibitory effects on target genes, resulting in the upregulation of these genes. For example, circRNA_000203 can regulate the occurrence of cardiac hypertrophy by directly sponging miR-26b-5p and miR-140-3p [23]. Apart from miRNA sponges, circRNAs can also function as protein sponges, such as RBP sponges. The binding of RBPs with circRNAs would lead to the function inhibition of RBPs, especially those participating in the transcription and translation of genes. For instance, circANRIL competitively recruited PES1 (pescadillo homolog 1, an essential 60S-preribosomal assembly factor), leading to the inhibition of ribosome biogenesis in vascular smooth muscle cells and macrophages [24]. CircANRIL induced nucleolar stress and p53 activation, which was followed by the induction of apoptosis and inhibition of proliferation in atherosclerosis. Moreover, in the presence of internal ribosomal enter sites and a corresponding open reading frame, circRNAs can affect protein translation and act as a scaffold for enzymes, guiding them to indicate location. CircFOXO3 binds to CDK2 and p21, contributing to the formation of the circFOXO3-p21–CDK2 ternary complex and then serving as a scaffold, affecting cancer cell-cycle progression [25]. In addition, circRNAs mediated the regulation of the transcription of parental genes. For example, Ci-ankrd52 can bind to the transcription sites and enhance host-gene transcriptional progress by acting as a positive regulator of Pol II transcription [26]. Although circRNAs were initially recognized as non-coding RNAs, studies conducted in recent years have demonstrated that circRNAs can serve as templates for protein translation via some modification [27]. CircZNF609 has an IRES element and can be translated into a protein that functions in myoblast proliferation [28].

## 3. Exploration of circRNAs

CircRNAs were first discovered in the early 1970s, however, due to limited available technology, they have been poorly studied in the past [29,30]. With the development of high-throughput sequencing (HTs) and bioinformatic tools, scientists have found that circRNAs are general features of the human transcriptome, and their biological functions have been intensively investigated. To date, there are more than 200,000 different circRNAs present in the union of all noncurated databases, to the best of our knowledge [31,32]. The current methods used to detect and quantify circRNAs include high-throughput sequencing (HTs), microarray and conventional RT-PCR/qPCR, and northern blot [33]. Most recently, Li et al. developed a rapid and useful screening tool for functional circular RNAs based on the CRISPR-Cas13d system [34]. This technology may provide a new tool for circRNA research. Cellular levels of circRNAs are known to be low in proliferating and neoplastic human cells. Recent studies performed on the human heart have shown that it expressed about 7000 to 16,000 different circRNAs [35]. In the cardiovascular system, circRNAs appear to be robustly expressed and show differential regulation in different related cells and cardiac diseases. Currently, there are several different datasets about circRNAs using microarray or HTs technology that focus on cardiovascular diseases such as atherosclerosis and acute myocardial infarction (MI). We summarized these gene expression omnibus (GEO) datasets in Table 2. This is a treasure to be discovered. Due to the space limitation, we would only focus on some cardiovascular-related cells and diseases. Data mining in the future may reveal more important circRNAs and their roles in CVDs.

## 4. circRNAs and Cardiovascular-Related Cells

There are many cells related to cardiovascular diseases. In this review, we focus on endothelial cells, smooth muscle cells, cardiomyocytes and cardiac fibroblast (Figure 2). Some other cells such as immune cells are not our primary focus here, but a special summary is also needed for these other cells in the future.

### 4.1. Endothelial Cells

Endothelial cells (ECs) are the foundation of the vascular system and EC injury occurs in the early stage of atherosclerosis. Several factors contribute to EC injury and dysfunction, such as oxidized low-density lipoprotein (ox-LDL), oxidative stress and hypoxia [70,71].

Ox-LDL is a common stimulating factor of atherosclerosis in vitro. Hsa_circ_0003575 was significantly upregulated in ox-LDL-stimulated human umbilical vein endothelial cells (HUVECs) in vitro. Functional tests indicated that silencing hsa_circ_0003575 could lead to the promotion of HUVEC proliferation and angiogenesis ability [72]. These results indicated that hsa_circ_0003575 may promote atherosclerosis by inducing cell apoptosis. In another study, circ_0003645 had been identified to promote the development of ox-LDL-induced HUVEC injuries. In that study, propofol protected against the viability inhibition and apoptosis promotion of HUVECs by decreasing the circ_0003645 level. Mechanically, circ_0003645 could induce TRAF7 upregulation following propofol treatment through sponging miR-149-3p [73].

The redox imbalance of the ECs plays a causative role in a variety of cardiovascular diseases. CircANKRD12, derived from the junction of exon 2 and exon 8 of the ANKRD12 gene, was significantly upregulated in H_2_O_2_-treated ECs. In a network analysis performed for the identified circANKRD12, the p53 and Foxo pathways were proven to play a fundamental role in the oxidative stress response in many different systems. Further, the downregulation of circANKRD12 affected the redox imbalance response, suggesting the potential role of circANKRD12 in the protection of ECs against oxidative stress [74].

In an EC and hypoxia study, cZNF292 was found to be expressed in the ECs and induced by hypoxia. Moreover, the silencing of cZNF292 reduced the tube formation and spheroid sprouting of ECs in vitro. The circRNA cZNF292 exhibits proangiogenic activities in vitro, and this circRNA was involved in the regulation of EC function. No validated microRNA-binding sites for cZNF292 were detected, indicating that cZNF292 may not act as a microRNA sponge [75].

Furthermore, Chen et al. demonstrated that CircDLGAP4 was significantly decreased in HUVECs suffering ischemia/reperfusion (I/R) injury [76]. CircDLGAP4, which acts as an miR-143 sponge, promoted HECT domain E3 ubiquitin protein ligase 1 (HECTD1) expression. HECTD1 could inhibit the apoptosis and migration in ECs associated with endoplasmic reticulum (ER) stress. The study suggested an important role for circDLGAP4 and HECTD1 in ER dysfunction induced by I/R.

In addition, the circRNA-0024103/miR-363/MMP-10 axis was reported to regulate endothelial cells behaviors such as proliferation, apoptosis, migration and invasion [77].

### 4.2. Smooth Muscle Cells

Vascular smooth muscle cell (SMC) is the major component of the medial layer and could maintain intravascular pressure and blood perfusion through coordinating vascular relaxation and contraction. 

Zeng et al. demonstrated that overexpression of circMAP3K5 inhibited the proliferation of human coronary artery SMCs [78]. Loss of TET2 was found to downregulate the circMAP3K5-mediated antiproliferative effect on vascular SMCs in SMC-specific TET2 knockout mice. CircMAP3K5/miR-22-3p/TET2 was found to be the mechanism axis.

In addition, circ-SATB2 was reported to regulate the differentiation, proliferation, apoptosis, and migration of VSMCs through enhancing STIM1 expression [79].

The knockdown of circSOD2 was found to inhibit PDGF-BB-induced SMC proliferation. On the contrary, circSOD2 ectopic expression promoted SMC proliferation. CircSOD2 acted as a sponge for miR-206, leading to the upregulation of notch receptor 3(NORCH3) and NOTCH3 signaling [80]. CircSOD2 is thus regarded as a novel regulator that mediates SMC proliferation and neointima formation following vascular injury.

What is more, Sun et al. reported that circ_RUSC2/miR-661/spleen-associated tyrosine kinase (SYK) could contribute to VSMC proliferation, phenotypic modulation and migration [81]. 

Furthermore, hsa_circ_0001445has been suggested as an indicator of stable coronary artery disease (CAD). This circRNA is produced from the SWI/SNF-related matrix-associated actin-dependent regulator of the chromatin subfamily A member 5 (SMARCA5) locus, and its levels in the plasma may be a predictor of coronary artery atherosclerosis in suspected patients. Interestingly, the decreased secretion of hsa_circ_0001445 could be observed when the human coronary SMCs were exposed to atherogenic conditions in vitro [82].

### 4.3. Cardiomyocytes

CircRNAs are mentioned as powerful cardiac development regulators affecting cardiac regeneration. CircNfix was overexpressed in adult hearts in humans, rats, and mice compared to infants. Experiments in vitro and in vivo indicated that cardiomyocyte proliferation was promoted when circNfix was downregulated. Huang et al. demonstrated that super-enhancer-regulated circNfix could suppress Ybx1 ubiquitin-dependent degradation and increase miR-214 activity to inhibit cardiac regenerative repair and functional recovery after myocardial infarction (MI) [83]. In addition, the upregulation of circSNRK could also contribute to the reduction of apoptosis and cardiomyocytes proliferation. In the post-infarction area after acute myocardial infarction (AMI), circSNRK promoted cardiomyocytes regeneration by acting as a sponge for miR-103-3p and upregulating the expression of SNRK [84]. 

What is more, Zhou et al. reported that circRNA-68566 could participate in myocardial I/R injury by regulating the miR-6322/PARP2 signaling pathway [85]. MiR-6322 was proven to be a direct target of circRNA-68566. CircRNA-0068566 inhibited I/R injury through reducing oxidative stress and apoptosis via miR-6322. A study conducted by Zong et al. indicated that overexpressed circANXA2 could inhibit hypoxia/reoxygenation (H/R)-treated H9C2 cell proliferation. Moreover, further study demonstrated that circANXA2 could reverse the inhibition of myocardial proliferation and increasing cardiomyocyte apoptosis by acting as a sponge for miR-133 [86]. Luo et al. demonstrated that suppressing circPVT1 expression could prevent heart I/R injury in rats and improve cardiomyocyte viability by regulating the circPVT1/miR-125b/miR-200a axis [87].

It has been reported that the absence of circ-CBFB could offer cardiac protection against H/R-triggered cardiomyocyte injury through the miR-495-3p/VDAC1 axis, suggesting its potential role for acute myocardial infarction treatment [88].

Additionally, hypoxia treatment upregulated the expression of circHSPG2 in AC-16 cells (human cardiomyocyte). In this study, exposing AC-16 cells to hypoxia resulted in a reduction in cell viability and proliferation as well as the promotion of apoptosis. The progressions were diminished by the silence of circHSPG2 [89].

Furthermore, hsa_circ_0000848 was notably downregulated in hypoxia-induced cardiomyocytes [90]. The silence of hsa_circ_0000848 inhibited the proliferation while accelerating the apoptosis. This circRNA interacted with the ELAV-like RNA-binding protein 1 protein to stabilize SMAD family member 7 mRNA and affected the development of cardiomyocyte cells cultured under hypoxia.

### 4.4. Cardiac Fibroblast

The activation and phenotypical transition of cardiac fibroblasts (CFs) could contribute to cardiac fibrosis. It was proven that circBMP2K enhanced the regulatory effects of miR-455-3p on its downstream target gene, SUMO1, which led to the inhibition of TGF-β1 or Ang II and resulted in the activation and proliferation of CFs [91]. In addition, circPAN3 knockdown was reported to attenuate autophagy-mediated cardiac fibrosis after myocardial infarction via the miR-221/FoxO3/ATG7 axis [92]. In another study, circRNA_010567 was proven to be markedly upregulated in CFstreated with Ang II. CircRNA_010567 silencing could upregulate miR-141 and downregulate TGF-β1 expression, and it suppressed fibrosis-associated protein resection in CFs, including Col I, Col III and α-SMA, which suggested that circRNA_010567 played an regulatory role in CFs [93].

Moreover, circNFIB, which was identified as a miR-433 sponge, was downregulated in adult CFs after treated with TGF-β [94]. The overexpression of circNFIB could attenuate the pro-proliferative effects induced by the miR-433 mimic, while the inhibition of circNFIB exhibited opposite results. CircNFIB is thus regarded as critical for protection against cardiac fibrosis.

## 5. circRNAs in Cardiac Diseases

Recent research has demonstrated that the profile expression of circRNAs is associated with different types of cardiovascular diseases, such as coronary artery disease, cardiomyopathies, chronic heart failure, hypertension, atrial fibrillation, and so on [95,96,97].

### 5.1. Atherosclerosis and Coronary Artery Diseases

CircRNAs played an important role in atherosclerosis and coronary artery diseases [26]. The SNPs on chromosome 9p21.3 were revealed to be correlated with the severity of atherosclerosis by a genome-wide association study (GWAS) [98]. The antisense non-coding RNA at the INK4 locus (ANRIL) and the related circular ANRIL (circANRIL) are transcribed on chromosome 9p21. It was reported that circANRIL upregulation could inhibit the development of vascular disorders, especially coronary artery diseases [99]. As mentioned above, circANRIL bound to PES1, impairing exonuclease-mediated pre-rRNA processing and ribosome biogenesis and leading to p53 activation [36,100]. This, in turn, led to a subsequent apoptosis increase, proliferation decrease, and the migration of VSMCs and macrophages. In addition, circANRIL overexpression could promote EC apoptosis and exacerbate EC inflammation. Therefore, circANRIL may play a role in atherosclerosis and CAD development by inducing the apoptosis and inflammation of these atherosclerotic-associated cells.

In addition, cardiomyocyte apoptosis and necrosis were important features in AMI. The CircRNA CDR1AS was found in abundance in the hearts of mice [101]. CDR1AS was a sponge for miR-7. CDR1AS was shown to be pro-apoptotic in vitro, consistent with the anti-apoptotic role of miR-7. More importantly, the overexpression of CDR1AS in mouse hearts resulted in larger infarct sizes after AMI, which could be prevented by the overexpression of miR-7. Recently, Wang et al. revealed that novel circRNA, mitochondrial fission and apoptosis-related circRNA (MFACR) could regulate mitochondrial dynamics and apoptosis in the heart by targeting the miR-652-3p-MTP18 signaling axis [102].

Recently, the experiments conducted by Si et al. illustrated the important role of circHipk3 in the regeneration of the heart after AMI. The expression of circHipk3 has also been found to be increased in fetal and neonatal mice hearts. An important observation is that the inhibition of circHipk3 expression also leads to the inhibition of the proliferation of cardiomyocytes [103]. In addition, circHIPK3 acted as a sponge for miR-133a to promote connective tissue growth factor (CTGF) expression, activating endothelial cells and improving cellular function. The overexpression of circHipk3 is associated with a decrease in cardiac dysfunction, which translates into a reduction in the area of fibrosis after AMI.

CiRS-7 was reported as a classic miRNA sponge. It has been confirmed that ciRS-7 has over 70 miR-7 binding sites [17,20]. Geng et al. reported increased ciRS-7 expression after MI in the cardiac tissue. In addition, when the ciRS-7 level was increased by a lentiviral-based overexpression in a rodent MI model, an increase in the extent of the MI area was also observed. The authors have also stated that ciRS-7 affected the axis of PARP/SP1 by sponging miR-7, and it thus regulated the apoptotic pathway.

Moreover, it is well known that plaque instability is very important in the pathogenetic mechanism of CAD. CircRNAs play a vital role in sustaining atherosclerotic plaque stability [104]. The results of the research obtained by Bazan et al. show the upregulation of circRNA-16 accompanying the downregulation of miR-221 in acutely ruptured carotid plaques. E26 transformation-specific-1 (ETS1), a key transcription factor of endothelial inflammation and tube formation, is the target of miR-221 [105]. MiR-221 bound to ETS1 and downregulated several EC inflammatory molecules and decreased the adherence of Jurkat T cells to activated HUVECs. In another study, miR-221-3p could promote pulmonary arterial SMC proliferation by targeting axis inhibition protein 2 (AXIN2) [106]. Therefore, circRNA-16 may play an important regulatory role in the stability of atherosclerotic plaques through acting as a sponge for miR-221.

In myocardial ischemia-reperfusion injury (MIRI) and the hypoxia/reoxygenation treatment models, the expressions of circ-GTF2I were significantly upregulated in vivo and in vitro when compared with that in the sham group. The knockdown of circ-GTF2I relieved neonatal rat cardiomyocyte damage and MI severity. Further study verified that circ-GTF2I induced the abnormal expressions of IL-6, TNF-α, LDH, Bax, Bcl-2, and Cyt-c in MIRI and the hypoxia/reoxygenation treatment models by regulating miR-590-5p and the heart development transcription factor KBTBD7. Circ-GTF2I promoted MIRI deterioration and induced the neonatal rat cardiomyocyte damage by targeting miR-590-5p and KBTBD7.

Transient receptor potential melastatin-3 (TRPM3, a calcium-permeable ion channel) is detected in VSMCs and is functionally related to contractility and the secretion of inflammatory factors such as IL-6 [107]. Nine kinds of circRNAs, including hsa_circ_0089378, could promote the expression of TRPM3 via interacting with hsa-miR-130a-3p in CAD patients [108]. This suggested that the progression of CAD may be regulated through the circRNA-miR-130a-3p/TRPM3 axis [109]. 

As stated above, different types of circRNAs have also been found to affect the function of atherosclerotic cells and plaque stability and participate in the development of CAD.

### 5.2. Cardiomyopathy and Heart Failure

Cardiomyopathies were recognized as a heterogeneous group of disorders of the myocardium that can change cardiac function (mechanical and/or electrical dysfunction) and structure and lead to heart failure. Heart failure (HF) represents one of the major challenges facing healthcare systems in industrialized societies, and an increasing burden in developing countries.

The circRNA exhibiting the highest level in the human heart is encoded by the SLC8A1 (solute carrier family 8 member A1) gene. Upregulated circSLC8A1 sequestered miR-133a to increase the expression of multiple miR-133a target genes, which indicated that the circSLC8A1/miR-133a-mRNAs axis may serve as a pivotal mechanism in cardiac hypertrophy pathogenesis [110]. In line with these observations, when compared with the control group, circSLC8A1 expression was elevated in the autopsy heart samples from sudden-cardiac-death patients with acute ischemic heart disease [111]. CircFndc3b is another significantly downregulated circRNA in the cardiac tissues of ischemic cardiomyopathy patients [47]. It interacts with the FUS RNA binding protein and increases vascular endothelial growth factor (VEGF)-A expression. This regulation enhances angiogenic activity and reduces cardiac endothelial cell apoptosis. In the hypoxic myocardium, the presence of circFndc3b in cardiac endothelial cells enhanced the function of the endothelial cells and protected cardiomyocytes against death.

cTTN1 is an abundant circRNA in the human heart and is downregulated in DCM [112]. RBM20 (RNA-binding motif protein 20) plays a critical role in the splicing of many cardiac genes, whose mutation will cause aggressive DCM [113,114]. RBM20 is dependent on cTTN and targets multiple key cardiac genes, such as calcium/calmodulin-dependent kinase II (CAMK2D) [115].

Another circRNA that regulates cardiac function is circ-Foxo3, which is usually increased in aged hearts [116]. In vitro, the ectopic expression of circ-Foxo3 induced senescence in fibroblasts; on the other hand, in vivo, the silencing of circ-Foxo3 reduced doxorubicin-induced cardiomyopathy in mice. Functionally, circ-Foxo3 could bind to several proteins that were involved in cellular stress response, including E2F transcription factor 1, inhibitor of DNA binding 1, focal adhesion kinase, and hypoxia-inducible factor 1α, resulting in the cytoplasmic sequestration of these proteins. Whether circ-Foxo3 contributes to cardiac ageing remains to be further investigated. 

Moreover, the circRNA described to be functional in the heart was termed heart-related circRNA (HRCR) [117]. Wang et al. conducted a study in mice. In this study, Wang et al. demonstrated that HRCR was normally expressed in mouse hearts and was repressed in hypertrophic and failing hearts. Biologically, HRCR seemed to function as an miRNA sponge, binding and thereby sequestering miR-223, an miRNA that caused cardiac hypertrophy via the inhibition of the protein ARC (apoptosis inhibitor with CARD domain). The overexpression of HRCR in an isoproterenol-induced hypertrophy mouse model inhibited hypertrophy, which the authors attributed to the inhibition of miR-223.

In HF tissues and H9C2 cells treated with oxygen–glucose deprivation (OGD), circSnap47 was upregulated when compared to the control group [118]. Wang et al. revealed that circSnap47 could relieve OGD-induced H9C2 cell damage and affect the progression of HF by inactivating the miR-223-3p/MAPK axis.

### 5.3. Hypertension

Essential hypertension is a multifactorial disease with high morbidity. A recent study found that, when compared to the healthy group, hsa-circ-0037909 was significantly upregulated in essential hypertension patients. This circRNA contributed to the pathogenesis of hypertension by acting as a sponge to inhibit miR-637 activity [119].

Hsa-circ-0005870 exhibited significant downregulation in patients with high blood pressure. Then, a network of hsa-circ-0005870-targeted miRNAs, including hsa-miR-5095, hsa-miR-1273g-3p, hsa-miR-6807-3p, hsa-miR-619-5p, and hsa-miR-5096, and their corresponding mRNAs was observed [55]. Hsa-circ-0005870 may represent a novel biomarker and the hsa-circ-0005870-miRNA-mRNA network may provide a potential mechanism for hypertension. 

### 5.4. Atrial Fibrillation

Atrial fibrillation (AF) is an abnormal heart rhythm characterized by the rapid and irregular beating of the atria. Zhang et al. [119] performed an association analysis of the AF-related circRNAs and their parental genes and revealed that hsa_circ_0000075 and hsa_circ_0082096 participated in the AF pathogenesis via the TGF-beta signaling pathway. 

In addition, circRNA calmodulin binding transcription activator 1 (circCAMTA1) was reported to be related to AF development [120]. CircCAMTA1 knockdown alleviated atrial fibrosis through downregulating TGFBR1 expression intermediated by miR-214-3p in AF.

### 5.5. Other Cardiovascular Diseases

Aortic dissection is an emergency and serious aneurysm disease in the cardiovascular system. Zheng et al. found an obviously upregulated circRNA in aortic tissues, hsa_circ_000595, from patients with aortic dissection aneurysms [121]. Hsa_circ_000595 was reported to promote the apoptosis of vascular smooth muscle cells (VSMCs) through upregulating miR-19a expression.

Vascular calcification (VC) is characterized by calcium phosphate crystals accumulating in the vessel wall. It is critical to reveal the novel mechanisms involved in VC as the pathogenesis is diverse and so many factors and mechanisms are involved. It was reported that circSamd4a had an anti-calcification property via the sponging of miR-125a-3p and miR-483-5p [122].

### 5.6. Predictor and Biomarker

Nowadays, a variety of circulating molecules, such as troponins, creatine kinase-MB and N-terminal pro brain natriuretic peptide (NT-proBNP), have been widely used in clinical laboratory tests. However, these molecules are easily influenced by factors such as age, medications and heart-associated diseases [123,124,125]. Circular RNAs have great biomarker potential for the following reasons: (1) they are extraordinarily stable due to the lack of exposed terminal ends [126]; (2) they have a large amount of cell-specific circRNA [127]; and (3) they are abundant in whole blood, plasma and extracellular vesicles [128]. Many studies have demonstrated the potential of circulating circRNAs as promising predictors and biomarkers. MICRA (myocardial infarction-related circular RNA) improves risk classification after MI [129]. Vausort et al. demonstrated reduced MICRA expression in MI patients and found that a lower MICRA level was related to a higher left ventricular dysfunction risk [130]. Further study found a close link between hsa-circRNA11783-2 and CAD in CAD patients’ peripheral blood by microarray. What is more, hsa_circ_0000284, hsa_circ_0001946, and hsa_circ_0008507 were found to be independent risk factors for CAD [131]. A recent study determined that hsa_circ_0124644 was closely associated with CAD, which could be used as a potential diagnostic biomarker for CAD, with a specificity of 0.626 and sensitivity of 0.861 [43]. Moreover, Wang et al. found that hsa_circ_0001879 and hsa_circ_0004104 were significantly upregulated in CAD patients compared with controls in another study [41]. As these circRNAs have been shown to have high sensitivity and specificity to CAD, they may be potential biomarkers of CAD. AF is a common complication in patients who have undergone coronary artery bypass grafting (CABG). In the plasma of patients with new-onset AF after isolated off-pump CABG, hsa_circ_025016 was found to be upregulated [132]. ROC analysis revealed a high diagnostic value, and it was confirmed by a large validation cohort. Another study has shown that, when compared with healthy controls, the expression of hsa_circ_0037911 in essential hypertension patients was upregulated. Hsa_circ_0037911 was proposed as a key circRNA for essential hypertension development, by affecting serum creatinine concentration, and a marker for the early detection of essential hypertension [133].

## 6. Conclusions and Future Direction

CVDs are the leading cause of death worldwide. In this review, we summarized the circRNAs involved in cardiovascular-related cells and diseases. Presently, the differential expression of circRNA in cardiovascular diseases has been observed, indicating that circRNAs might participate in the pathophysiological processes of diseases and could be used as biomarkers and therapeutic targets for disease. However, circRNA research in CVDs is still in its infancy and there is still a long way to go.

Firstly, the lack of a uniform and standard naming system and detection method for circRNAs may confuse the researchers and be a hindrance for communications among different laboratories. Secondly, though there has been technical progress in the past decade for circRNAs, it is still difficult to verify the roles of circRNAs in vivo by overexpression or downregulation, and most functional studies have focused on the sponge as it is relatively easy to be validated. More researches are needed in the future to reveal the functions of circRNAs. Moreover, the mechanisms for circRNAs in different diseases, especially CVDs, need to be further clarified. Thirdly, circRNAs are more stable than linear RNAs and are detectable in body fluids such as peripheral blood through exosome secreting, making them potential biomarkers for cardiovascular diseases. However, these biomarkers need to be further verified by more different laboratories, and several conditions, such as sensitivity, specificity, feasibility, reliability, and repeatability, need to be optimized to meet the clinical test criteria. Finally, progress in the circRNA field might also expand their therapeutic potential. The high stability of circRNAs makes them potential long-lasting regulators of specific cellular functions. In a recent study, the overexpression of an artificial circRNA could inhibit HCV viral protein production through sponging the liver-specific miRNA-122, which is required for the life cycle of the hepatitis C virus (HCV) [134]. As a rising research star, circRNAs have potential in therapies such as circRNAs vaccines and genetically edited treatments.

In brief, we have illustrated the landscape of circRNAs in cardiovascular diseases and shed light on the importance and potential effectiveness of circRNAs in the diagnosis and therapy of CVDs.

## Figures and Tables

**Figure 1 ijms-24-04571-f001:**
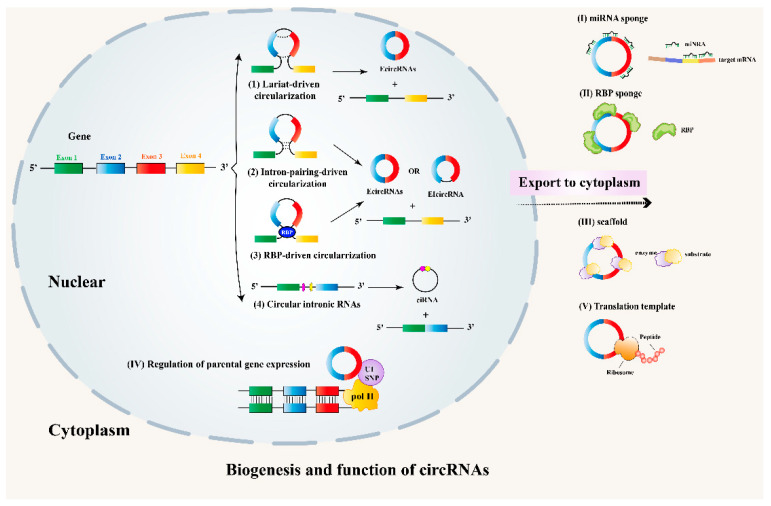
Biogenesis and functions of circRNAs. Classical mechanisms for circRNAs generation: (1) Exon-skipping or lariat-driven circularization, (2) Intron-pairing-driven circularization, (3) RBP-driven circularization, (4) Special lariat circularization for CiRNA. The five representative biological functions of circRNAs: miRNA sponge, interacting with RBP, scaffold for modulating protein–protein interactions, regulation of parental gene expression and translation template. EcircRNAs: exonic circRNA; EIcircRNAs: exon-intronic circRNA; ciRNA: circular intronic RNA.

**Figure 2 ijms-24-04571-f002:**
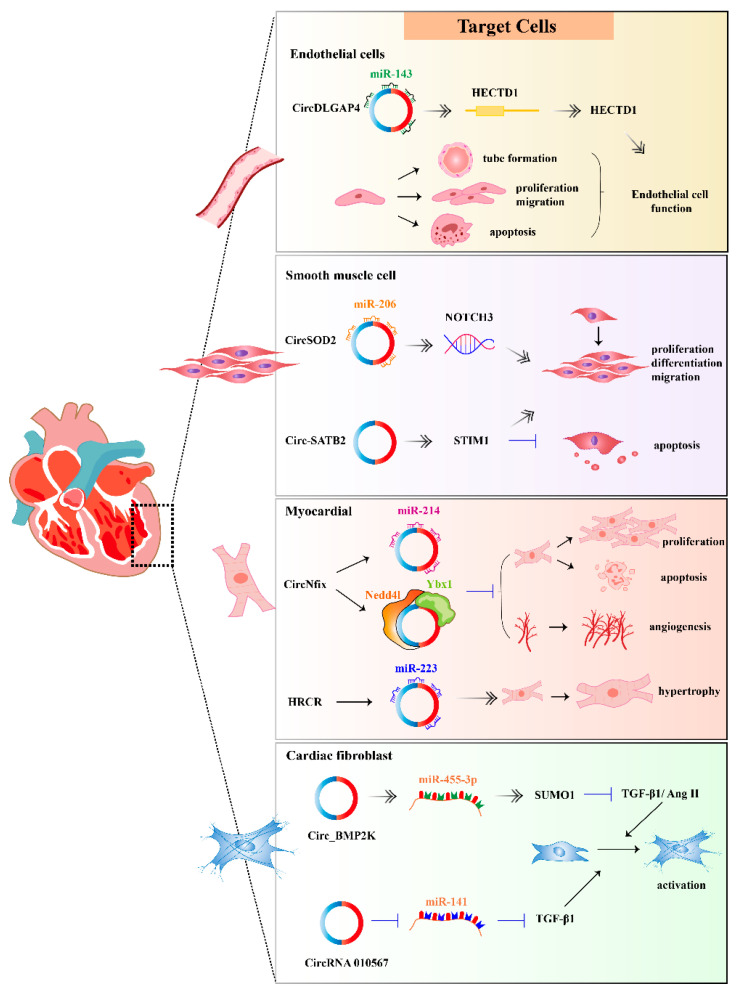
The typical mechanism diagram for circRNAs in cardiovascular targets cells. HECTD1: HECT domain E3 ubiquitin protein ligase 1; NOTCH3: notch receptor 3; STIM1: stromal interaction molecule 1.

**Table 1 ijms-24-04571-t001:** Classification and differences of circRNAs.

Category	EcircRNAs	EIcircRNAs	CiRNAs
Origin	Exons	Exons and introns	Introns
Location	Mostly cytoplasm	Nucleus	Nucleus
Formation mechanisms	Lariat-driven circularization; Intron-pairing-driven circularization; RBP-driven circularization	Intron-pairing-driven circularization; RBP-driven circularization	Special lariat-driven circularization

**Table 2 ijms-24-04571-t002:** Recent identification of circRNAs in CVDs from gene expression omnibus (GEO) datasets. AS: atherosclerosis; VSMCs: vascular smooth muscle cells; CAD: coronary artery disease; AMI: acute myocardial infarction; I/R: ischemia/reperfusion; DCM: dilated cardiomyopathy; HCM: hypertrophic cardiomyopathy; RVHD, rheumatic valvular heart disease; CAVD: calcified aortic valve disease; HTs: high-throughput sequencing.

GEO Number	Disease	Sample	Method	Related circRNAs	Main Findings	Ref.
GSE216305	Atherosclerosis	Homo sapiens (VSMCs)	microarray	circCTDP1	VSMC within atherosclerotic plaques mediates the link between glycolysis switching and phenotype transformation via KLF4-eEF1A2/circCTDP1-PFKFB3 axis	[36]
GSE133270	Atherosclerosis	Mouse AS model (VSMCs)	HTs	circEsyt2	circEsyt2 regulates vascular remodeling by regulating p53-beta splicing	[37]
GSE151475	Atherosclerosis	Homo sapiens (HUVEC/HCAEC)	HTs	circGNAQ (hsa_circ_0006459)	The circGNAQ/miR-146a-5p/PLK2 axis is used to regulate endothelial cell aging and atherosclerosis progression	[38]
GSE107522	Atherosclerosis	Homo sapiens (macrophages)	microarray	hsa_circ_0007478	The hsa_circ_0007478/miR-765/EFNA3 axis regulates lipid metabolism and foam cell formation in macrophages, thus participating in the regulation of vascular atherosclerosis	[39]
GSE65392	Atherosclerosis	Homo sapiens HEK293 cell line	microarray	circANRIL	CircANRIL and PES1 induce nucleolar stress and p53 activation to regulate atherosclerosis	[24]
GSE208194	CAD	Homo sapiens (plasma)	HTs	circUBAC2	Circulatory levels of circUBAC2 were higher expressed in patients with myocardial infarction than in healthy controls	[40]
GSE115733	CAD	Homo sapiens (PBMC)	microarray	Hsa_circ_0001879 and Hsa_circ_0004104	hsa_circ_0001879 and hsa_circ_0004104 can be used as new biomarkers to diagnose CAD	[41]
GSE152498	CAD	Homo sapiens (PBMC)	HTs	hsa_circ_0005540	Plasma exosomal hsa_circ_0005540 can be used as a promising diagnostic biomarker of CAD	[42]
/	CAD	Homo sapiens (PBMC)	microarray	hsa_circ_0124644	hsa_circ_0124644 can be used as a diagnostic biomarker of CAD	[43]
GSE149051	AMI	Homo sapiens (Blood)	microarray	top three upregulated: circRNA: hsa_circ_0050908, hsa_circRNA4010-22, hsa_circ_0081241; top three downregulated: hsa_circ_0066439, hsa_circ_0054211, hsa_circ_0095920	circRNA may be involved in the pathogenesis of AMI (such as hsa_circ_0050908, hsa_circRNA4010-22, hsa_circ_0081241…)	[44]
GSE169594	AMI	Homo sapiens (Blood)	microarray	circRNA_104761	circRNA_104761 can be used as a diagnostic marker of AMI and distinguish the severity of coronary lesions	[45]
GSE160717	AMI	Homo sapiens (Blood)	microarray	hsa_circRNA_001654, hsa_circRNA_091761, hsa_circRNA_405624, hsa_circRNA_406698	Four CircRNAs (hsa_circRNA_001654, hsa_circRNA_091761, hsa_circRNA_405624, and hsa_circRNA_406698) modulate the activation and expression of RUNX1 in AMI patients via miRNA sponges	[46]
GSE133503	AMI	Mus musculus (heart)	microarray	circFndc3b	Cardiac repair after myocardial infarction is regulated by the CircFndc3b/FUS/VEGF-A axis	[47]
/	Myocardial ischemia reperfusion	Mouse model of I/R injury	HTs	mmu_circRNA_0001379, mmu_circRNA_0002263	Nineteen upregulated and 20 downregulated circRNAs were identified to be involved in differential expression in myocardial I/R injury	[48]
/	Myocardial ischemia reperfusion	HCM cells simulated with myocardial I/R	/	circHIPK3	CircHIPK3 sponge miRNA-124-3p inhibits myocardial cell proliferation and induces apoptosis after I/R injury	[49]
/	Myocardial ischemia reperfusion	Cardiomyocytes/cardiac tissues	/	circ-NNT	The circ-NNT/miR-33a-5p/USP46 signal axis was used to promote pyroptosis and myocardial I/R injury	[50]
/	Myocardial ischemia reperfusion	Mouse model of I/R injury	/	circ_Ddx60	Cardiomyocyte apoptosis was inhibited by regulating the circ_Ddx60/miR-302a-3p/Bcl2a1a axis	[51]
/	Myocardial ischemia reperfusion	I/R rat model and hypoxia/re-oxygenation (H/R)-treated H9C2 cells	/	circRNA_0031672	The circRNA_0031672/miR-21-5p/PDCD4 signaling pathway mediated the apoptosis of cardiomyocytes and alleviated the IRI of cardiomyocytes	[52]
/	Myocardial ischemia reperfusion	Myocardial I/R model in vitro by oxygen and glucose deprivation and reperfusion in cardiomyocytes	/	circ_0050908	Prevention of myocardial I/R injury by the Circ_0050908/miR-324-5p/TRAF3 axis	[53]
/	Myocardial ischemia reperfusion	Extracellular vesicles (EVs)	HTs	mmu-circ008351, mmu-circ001007, mmu-circ008228, mmu_circ_0001336 and mmu-circ007845	In the I/R group (such as mmu-circ008351, mmu-circ001007, mmu-circ008228…), 185 significantly differentially expressed (DE) circrnas were identified in cEVs	[54]
/	Hypertension	Homo sapiens	microarray	has-circ-0000437, has-circ-0008139, has-circ-0005870, has-circ-0040809	hsa-circ-0005870 could serve as a biomarker for hypertension diagnosis	[55]
/	Essential hypertension (EH)	Homo sapiens	microarray	hsa_circ_0105015	hsa_circ_0105015 combined with hsa-miR-637 indicates vascular inflammation or endothelial dysfunction and is a biomarker for early diagnosis of EH	[56]
GSE134584	Heart failure (Plasma)	Homo sapiens (plasma)	microarray	hsa_circ_0062960	hsa_circ_0062960 may be involved in the platelet activity of HF and potentially used to predict prognosis	[57]
GSE162505	DCM	Homo sapiens (heart)	HTs	top3: Chr7:8257935−8275635, chr4: 187627717−187630999−, chr1: 219352489−219385095+	In the DCM group, 213 circRNAs and 617 mRNAs were identified as significantly upregulated. 85 circRNAs and 1125 mRNAs were significantly downregulated	[58]
GSE148602	HCM	Homo sapiens (blood)	microarray	hsa_circ_0043762, hsa_circ_0036248 and hsa_circ_0071269	hsa_circ_0043762, hsa_circ_0036248 and hsa_circ_0071269 may be involved in the risk factors of HCM.	[59]
GSE122905	Idiopathic constrictive pericarditis	Homo sapiens (pericardium)	HTs	hsa_circ_0008679, hsa_circ_0006238, hsa_circ_0013093	hsa_circ_0008679, hsa_circ_0006238 and hsa_circ_0013093 were identified to be differentially expressed in CP	[60]
GSE129409	Atrial fibrillation	Homo sapiens (left atrial appendage)	microarray	has_circRNA_100612	The potential roles of has_circRNA_100612, has-miR-133b, and KCNIP1/JPH2/ADRB1 in atrial fibrillation	[61]
GSE197764	Cardiac arrest	Homo sapiens (blood)	HTs	circNFAT5	circNFAT5 was used to predict clinical outcome after cardiac arrest	[62]
GSE97745	Thoracic aortic dissection	Homo sapiens (aortic specimens)	microarray	hsa_circRNA_101238, hsa_circRNA_104634, hsa_circRNA_002271, hsa_circRNA_102771, hsa_circRNA_104349	The hsa_circRNA_101238/hsa-miR-320a/MMP9 signal axis was involved in the regulation of aortic dissection	[63]
GSE215935	Aortic dissection	Mus musculus ()	microarray	mmu_circ_0004377, mmu_circ_0004375, mmu_circ_0004373, mmu_circ_0004371, mmu_circ_0004370	Differential circRNAs were identified in OSA-AD animal models (such as mmu_circ_0004377, mmu_circ_0004375, mmu_circ_0004373…)	[64]
GSE171827	Pulmonary hypertension secondary to congenital heart disease	Homo sapiens	microarray	circ_003416 downregulated, circ_005372 upregulated	Circ_003416 and circ_005372 are involved in oxidative phosphorylation and tight signaling to regulate pulmonary hypertension	[65]
GSE145610	Tetralogy of Fallot	Homo sapiens (heart)	microarray	hsa_circ_0007798	The hsa_circ_0007798/miR-199b-5p/hf1a signaling axis is involved as a risk factor for TOF	[66]
GSE168932	Rheumatic valvular heart disease (RVHD)	Homo sapiens (plasma)	microarray	Has_circ_0000437	Has_circ_0000437 can promote the process of RVHD and may be a potential for the diagnosis and treatment of RVHD	[67]
GSE155119	CAVD	Homo sapiens (valve leaflets)	microarray	circ-CCND1	The circ-CCND1/miR-138-5p/CCND1/P53/P21 pathway is involved in the regulation of the development of CAVD	[68]
GSE144431	Abdominal aortic aneurysm	Homo sapiens	microarray	hsa (Homo sapiens) _circ_0005360 (LDLR) and hsa_circ_0002168 (TMEM189)	The hsa_circ_0005360/miR-181b and hsa_circ_0002168/miR-15a axis may play a regulatory role in the occurrence and development of human abdominal aortic aneurysm	[69]

## Data Availability

Not applicable.

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
