# Peer review of "The Landscape of Circular RNAs in Cardiovascular Diseases"

_ijms, 2023, doi:10.3390/ijms24054571_

Round 1

Reviewer 1 Report

Dear Editor,

The topic of the article about circRNAs is very interesting. They could be biomarkers in cardiovascular diseases considering that they are stable in biological fluids.

However, the article does not fully develop the importance, function and classification of these RNAs. The figures describing their biogenesis are extremely similar to those in the article below:

Zhang L, Zhang Y, Wang Y, Zhao Y, Ding H, Li P. Circular RNAs: Functions and Clinical Significance in Cardiovascular Disease. Front Cell Dev Biol. 2020 

It would be best to redo it and add a table with their classification and functions

Author Response

Thanks for your bluntly and constructive comment. We have redrawn Figure 1 by integrating panel A and panel B. We also added this article into the Reference now. A new Table about classification and differences of circRNAs was now added and named Table 1.

Reviewer 2 Report

The reviewed manuscript entitled ‘The landscape of circular RNAs in cardiovascular diseases’ written by Qi Long et al. gathers information about the biogenesis of circRNAs and their role in cardiovascular diseases, including atherosclerosis, CAD, heart failure, cardiomyopathy, hypertension, and others. The review describes circRNAs as important factors involved in pathophysiological processes that contribute to CVD development. The text of this manuscript is not high quality, but understandable and clear. The references cited are mostly recent publications and relevant. However, some missing points exist and should be filled in. The specific comments I provided below.

Major comments:

1    1. In the literature, there are several similar review articles on this topic in recent years (https://doi.org/10.1007/s11010-021-04286-z, 10.1097/FJC.0000000000000841, https://doi.org/10.1016/j.omtn.2020.06.022, https://doi.org/10.1016/j.phrs.2021.105766, https://doi.org/10.1016/j.phrs.2021.105766, https://doi.org/10.3892/ijmm.2020.4792, https://doi.org/10.3389/fcell.2020.584051, https://doi.org/10.1002/jcp.27384, https://doi.org/10.1111/jcmm.16203). Please refer to these articles to highlight need for the current review and to make it relevant and of interest to the scientific community.

2   2. Short paragraph containing methodological aspects of the reviewed work, including searched libraries and used keywords, should be added, maybe as last paragraph of the Introduction section.

3   3. In panel B of Figure 1 the ciRNA is presented, but not appeared in panel A. It could confuse the readers. Please add the biogenesis mechanism of ciRNAs to panel A of Figure 1.

4   4. Table 1 was mentioned in line 112, but it was provided. Please add Table 1.

Minor comments:

1   1. Figure 2 was not referenced in the text.

2   2. Figure 1B – miRNA should be instead of miNRA.

3   3. The 4.5 paragraph should have number 5.

4   4. There are some minor text errors, especially check lines 77, 173, 191, 208, 325, and 348. 

I believe that my suggestions will be helpful to the authors in increasing the quality of the reviewed manuscript.

Round 2

Reviewer 1 Report

The requested changes have been made

Reviewer 2 Report

I thank the authors for taking my suggestions into consideration. I have no further comments.